# Coenzyme Q10: Role in Less Common Age-Related Disorders

**DOI:** 10.3390/antiox11112293

**Published:** 2022-11-19

**Authors:** David Mantle, Iain P. Hargreaves

**Affiliations:** 1Pharma Nord (UK) Ltd., Telford Court, Morpeth, Northumberland NE61 2DB, UK; 2School of Pharmacy, Liverpool John Moores University, Liverpool L3 3AF, UK

**Keywords:** coenzyme Q10, age related disorders, late onset disorders, multiple system atrophy, progressive supranuclear palsy, sporadic adult onset ataxia

## Abstract

In this article we have reviewed the potential role of coenzyme Q10 (CoQ10) in the pathogenesis and treatment of a number of less common age-related disorders, for many of which effective therapies are not currently available. For most of these disorders, mitochondrial dysfunction, oxidative stress and inflammation have been implicated in the disease process, providing a rationale for the potential therapeutic use of CoQ10, because of its key roles in mitochondrial function, as an antioxidant, and as an anti-inflammatory agent. Disorders reviewed in the article include multi system atrophy, progressive supranuclear palsy, sporadic adult onset ataxia, and pulmonary fibrosis, together with late onset versions of Huntington’s disease, Alexander disease, lupus, anti-phospholipid syndrome, lysosomal storage disorders, fibromyalgia, Machado-Joseph disease, acyl-CoA dehydrogenase deficiency, and Leber’s optic neuropathy.

## 1. Introduction

Coenzyme Q10 (CoQ10) is usually described as a vitamin-like substance, although it is endogenously synthesised within most cell types. CoQ10 has a number of functions of key importance to normal cell function; these include: (i) Its key role in cellular energy supply via mitochondrial oxidative phosphorylation; (ii) Its role as a major endogenously synthesised lipid soluble antioxidant, protecting cellular/subcellular organelle membranes from free radical induced oxidative damage; (iii) Its role in the metabolism of lysosomes, sulphides, amino acids and cholesterol; (iv) Its role in directly mediating the expression of more than one hundred genes, including those involved in the inflammatory process [1,2]. Because of its multiplicity of roles in cell function, it is not surprising that a deficiency of CoQ10 has been implicated in the pathogenesis of a wide range of disorders. The role of coenzyme Q10 (CoQ10) in some of the more common disorders has received considerable attention; for example there have been more than 700 articles (including 97 randomised controlled trials) relating to CoQ10 and heart disease in man published in the peer-reviewed medical literature (as listed on Medline). Similarly, there have been almost 300 articles published on CoQ10 and diabetes and CoQ10 and Parkinson’s disease, respectively. However, the role of CoQ10 in some of the less common disorders has received somewhat less attention; in this article we have therefore reviewed the potential role of CoQ10 in the pathogenesis of less common disorders such as multiple system atrophy (MSA), progressive supranuclear palsy (PSP), sporadic adult onset ataxia, and pulmonary fibrosis, together with late onset versions of Huntington’s disease, Alexander disease, lupus, anti-phospholipid syndrome, lysosomal storage disorders, fibromyalgia, Machado-Joseph disease, acyl-CoA dehydrogenase deficiency, and Leber’s optic neuropathy.

## 2. Multiple System Atrophy

Multiple system atrophy (MSA) is a neurodegenerative disorder resulting from progressive degeneration of neurons and glia which is characterized by a variable combination of autonomic impairment, Parkinsonism and ataxia [3]. The age of onset of MSA is generally around 55–60 years, with an annual incidence of 3/100,000 in those aged 50 years and above [4]. The administration of the pharmaceutical Paroxetine, which is a selective serotonin reuptake inhibitor, has been reported to provide some clinical benefits to some patients with MSA [5]; otherwise, there is no current treatment option available that can delay the disease progression of MSA, which has a mean survival time of approximately 9 years [6]. MSA is a member of a group of disorders known as synucleinopathies which are characterised by the deposition of abnormal mis-folded alpha-synuclein protein in both the central and peripheral nervous system [7]. Histologically MSA can be subdivided into Parkinsonian (MSA-P) and cerebellar phenotypes (MSA-C) by selective atrophy and neuronal loss in striatonigral and olivopontocerebellar systems [8]. The main pathological hallmark of MSA is the appearance of cytoplasmic inclusions in oligodendrocytes [9]. Biochemically, MSA is characterised by mitochondrial dysfunction [10], oxidative stress [11], inflammation [12], and depletion of CoQ10, as described below.

A number of clinical studies have reported significantly decreased levels of CoQ10 levels in plasma, cerebrospinal fluid (CSF) and the cerebellar tissue of patients with MSA. The deficit in CoQ10 status has been associated with abnormalities in the *COQ2* gene which encodes the enzyme, coenzyme-Q2-polyprenyltransferase, which catalysis the condensation of the benzoquinone nucleus of CoQ10 with the isoprenoid side chain. However, not all MSA patients have an aberrant *COQ2* gene [13]. Furthermore, mutations in the genes that encode other enzymes in the CoQ10 biosynthetic pathway have also been reported, including, *PDSS1* (Decaprenyl-diphosphate synthase subunit 1), *COQ5* (methyltransferase), and *COQ7* (5-demethoxyubiquinone hydroxylase) [14]. In a study by Mitsui et al. [15] which assessed the plasma CoQ10 status of 44 MSA patients, an approximate 30% decrease in the mean plasma CoQ10 level of this isoprenoid was reported compared to control levels, regardless of patient *COQ2* genotype. In a subsequent study by Kasai et al. [16] a significant decrease in serum CoQ10 levels (when related total circulatory cholesterol level) was reported in 18 MSA patients (*COQ2* genotype not specified) compared to control. Total circulatory cholesterol acts as a marker of the lipoprotein fraction of the blood, the main carriers of CoQ10 in the circulation. In a cohort of 40 MSA patients (COQ2 phenotype not specified), Du et al. [17] reported a significant decrease in plasma CoQ10 levels when compared to levels in the controls. In a study by Compta et al. [18] a significant decrease in the level of CSF CoQ10 was reported in 20 MSA patients (COQ2 genotype not specified) in comparison to that of patients with related neurological disorders (Parkinson’s disease, progressive supranuclear palsy) or controls.

With regard to the brain, CioQ10 levels are reported to be lowest within the cerebellum, and therefore, this brain region is thought to be more vulnerable to a CoQ10 deficiency. In a post mortem investigation by Barca et al. [19] CoQ10 levels were found to be significantly decreased by 40% in cerebellar tissue from a cohort of 12 MSA patients (without *COQ2* mutations) in comparison to control levels. Although none of the MSA patients had *COQ2* mutations, there were significant reductions in the protein level of the PDSS1 and COQ5 CoQ10 biosynthetic enzymes. The CoQ10 levels in both striatal and occipital cortical samples from these MSA patients were comparable to control levels [19]. In a subsequent study by, Schottlaender et al. [20] a significant decrease in post mortem cerebellum CoQ10 levels was reported in a series of 20 MSA patients, although the decrease in CoQ10 levels was 3–5% of the control tissue. Decreased ATP levels in association with reduced expression of the COQ2 and COQ7 CoQ10 biosynthetic enzymes have been reported in disease-affected brain regions (principally cerebellum, as well as putamen) of MSA patients [21]. Interestingly, cultured neurons from MSA patients appear to have a severe impairment in autophagy as indicated by reduced autophagic flux, as well as decreased α-mannosidase and β-mannosidase activities, together with increased basal autophagy rates [22]; in addition, one of the newly determined functions of CoQ10 is its involvement in maintenance lysosomal pH [23].

Assessment of CoQ10 supplementation in MSA has been restricted to studies in cell culture, or single patient case investigations. Using MSA patients neuronal cells with and without the *COQ2* mutation, Nakamoto et al. [24] identified evidence of cellular dysfunctions attributable to a decrease in CoQ10 status, which were in part resolved as the result of exogenous CoQ10 supplementation. In an MSA patient with the *COQ2* mutation and in an advanced stage of the disease, Mitsui et al. [25] reported high dose CoQ10 supplementation (1200 mg/day) caused an improvement in brain energy metabolism (as measured by cerebral oxygen metabolic rate/positron emission tomography). There appears to be a clear rationale for the involvement of CoQ10 in the pathogenesis of MSA, and a number of clinical studies have reported evidence of decreased CoQ10 levels in blood and brain tissue from MSA patients. At present, there has been no randomised controlled trial assessing the supplementary use of CoQ10 in the treatment of MSA, and this is clearly warranted. One issue which needs to be first resolved however is the optimal formulation to enable supplementary CoQ10 to cross the blood–brain barrier [26].

## 3. Progressive Supranuclear Palsy

Progressive supranuclear palsy (PSP) is a disorder resulting from the deposition of aggregated tau protein in brain tissue, resulting in problems with balance, movement and vision [27]. The incidence of this disorder is 5/100,000, with a mean age of onset of 63 years and mean survival time of 7 years [28]. There is currently no effective therapy to treat symptoms or slow progression of PSP. Biochemically, PSP is associated with mitochondrial dysfunction, oxidative stress and inflammation [29,30,31] and therefore, in view of its role in mitochondrial energy metabolism, its antioxidant function together with its its anti-inflammatory effect, CoQ10 may be an appropriate therapeutic agent for this disorder [1,2]. To date there have been two randomised controlled trials of supplementary CoQ10 in PSP. In the study by Stamelou et al. [32] of 20 PSP patients, supplementation with CoQ10 at a dosage of 5 mg/kg/day for 6 weeks resulted in improved cerebral energy metabolism assessed via magnetic resonance spectroscopy, as well as an improvement in PSP rating scale. In the study by Apetauerova et al. [33] of 60 PSP patients, supplementation with CoQ10 (2400 mg/day) for up to 12 months did not significantly improve PSP symptoms or disease progression; however the study had a high patient drop-out rate and lacked the precision to exclude a moderate benefit of CoQ10 supplementation.

## 4. Sporadic Adult Onset Ataxia

Sporadic adult-onset ataxia (SAOA) is a non-genetic neurodegenerative disorder of the cerebellum of as yet uncertain origin which presents with progressive ataxia. SAOA is distinguished from hereditary ataxias (such as late onset Friedreich’s ataxia) and from acquired ataxias (such as toxin, alcohol or infection induced ataxia) [34]. Compared with MSA, disease progression is significantly slower [35]. Prevalence rates are estimated to be in the range of 2 to 8/100,000, and the age of onset is typically after 50 years of age [36]. Neuropathological and imaging studies of SAOA patients have revealed isolated cerebellar cortical degeneration with mild to no detectale brainstem involvement. As both the aetiology and pathogenesis of SAOA are as yet unknown, there is no specific therapeutic treatment for this condition. At present there have been no clinical investigations to determine the potential role of CoQ10 in the pathogenesis or treatment of SAOA; given that the levels of CoQ10 in human brain are reported to be lowest in cerebellum, and that the latter tissue may be selectively vulnerable to CoQ10 deficiency, we would suggest that this is an area worthy of further investigation.

## 5. Pulmonary Fibrosis

Pulmonary fibrosis is an uncommon lung disorder that causes progressive irreversible scarring of the lungs, with poor prognosis [37]. The incidence is approximately 30/100,000, with a mean age of onset of 62 years and mean survival time of 3.5 years [38]. Fibrosis is the formation of fibrous connective tissue, particularly collagen, by activated fibroblasts. Fibrosis is a response to tissue injury, and is an integral part of the wound healing and tissue repair process. In the young this fibrous connective tissue is replaced over time by new functional tissue. However, in older people tissue scarring may persist and continue to form which can result eventually in the loss of organ function. Pulmonary fibrosis appears to be triggered by mitochondrial dysfunction and free radical induced oxidative stress, followed by subsequent fibrosis [39,40]. There are currently no effective treatments for this disorder, although stem cell transplant may provide a potential therapy. Fujimoto et al. [41] reported reduced serum CoQ10 levels in a small cohort of pulmonary fibrosis patients. To date there have been no randomised controlled trials of CoQ10 for pulmonary fibrosis. However, using cultured cells, Sugizaki et al. [42] were able to demonstrate that the CoQ10 analogue, idebenone was able to supress bleomycin induced activation of lung fibroblasts. In addition, Liu et al. [43] reported that CoQ10 treatment increased the efficacy of airway basal stem cell transplantation in bleomycin-induced idiopathic pulmonary fibrosis in mice.

## 6. Late Onset Huntington’s Disease

Huntington’s disease (HD) is a neurodegenerative disorder characterized by progressive motor decline, cognitive impairment and behavioural abnormalities. The disorder results from a cytosine-adenine-guanine (CAG) repeat expansion, on the 5′ end of the *Huntingtin* gene [44]. The prevalence of HD is approximately 7/100,000; death occurs within 15 to 20 years after the initial diagnosis, for which there is no effective therapy. The age of onset is typically in the range 30–50 years of age, but approximately 10% of patients develop late onset Huntington’s disease (LOHD) after the age of 60 [45]. Diagnosis of LOHD can be missed because of the perceived low likelihood of HD in the over 60-year-olds. Individuals with LOHD may have a slower progression of disease [45]. The repeat length of the CAG expansion is an important indicator for the age of onset of the disease [46]. Mitochondrial dysfunction, oxidative stress and inflammation are associated with the neurodegenerative process in HD [47,48,49]. Evidence of defective energy metabolism was provided by Koroshetz et al. [50]. In this study 31P magnetic resonance spectroscopy indicated evidence of a significant decrease in the phosphocreatine to inorganic phosphate ratio in resting muscle of HD patients. The CSF lactate-pyruvate ratio was also significantly increased in HD patients together with increased lactate concentrations (assessed using 1H nmr spectroscopy) which were detected in HD cerebral cortex. In view of its potential to enhance mitochondrial function, CoQ10 supplementation was investigated in HD patients and found to result in a significant decrease in the lactate concentrations in the cerebral cortex of the patients. Andrich et al. [51] reported reduced serum levels of CoQ10 in HD patients. Animal models of HD have shown symptomatic benefit following CoQ10 administration; for example, oral CoQ10 supplementation was found to significantly extend the survival rate and also delayed the development of motor deficits, weight loss, cerebral atrophy, and neuronal intranuclear inclusions in the R6/2 transgenic HD mouse model [52]. Similarly, dietary administration of CoQ10 (0.2%) improved behavioural deficits in the CAG140KI slow progression mouse model [53]. However, randomised controlled trials supplementing CoQ10, 600 mg/day for 30 months or 2400 mg/day for 60 months [54,55], failed to slow disease progression in HD patients. Similarly, a 1-year randomised controlled trial of the CoQ10 analogue idebenone in HD patients failed to slow progression of the disease [56]. To date, there have been no randomised controlled trials of CoQ10 specifically in LOHD patients, and this may be worthwhile given the slower disease course in the latter patients.

## 7. Late Onset Alexander Disease

Alexander disease is a neurodegenerative disorder affecting cerebral white matter, resulting from mutations in the protein glial acidic fibrillary protein (GFAP) in astrocytes [57]. The age of onset is typically in early life (neonatal/infantile/juvenile), although there is also a late onset form of Alexander disease (LOAD) [58]. The influence of polymorphism in the GFAP promotor on the age of disease onset is described by Yoshida et al. [59] The incidence of Alexander’s disease is not accurately known, but has been estimated as 1–2/million. There is no effective treatment for Alexander disease. Evidence from cell culture, animal models and clinical studies has implicated mitochondrial dysfunction, oxidative stress and inflammation in the disease process and therefore in view of its function within the cell, CoQ10 may be an appropriate therapeutic strategy to target these abnormalities [60,61]. To date there have been no clinical studies to supplement CoQ10 in Alexander disease.

## 8. Late Onset Lupus

Lupus is an autoimmune disorder that can affect the skin, joints, kidneys and other organs [62]. Lupus is typically diagnosed between the ages of 15 and 45, but approximately 10–15% of lupus patients are affected by a late onset form of the disease manifesting after the age of 50 [63]. Although late onset lupus is reported to have a milder course than general lupus, such patients do not present with typical SLE symptoms or serology, which can lead to a major delay in diagnosis. Treatment of late onset lupus may involve NSAIDs, corticosteroids and immunosuppressive agents, as well as the antimalarial agent hydroxychloroquin, all of which have associated adverse effects [63]. Lupus is associated with mitochondrial dysfunction, oxidative stress and inflammation [64,65,66] and therefore may be amenable to CoQ10 therapy [1,2]. To date there have been no randomised controlled trials of supplemental CoQ10 in late onset or general forms of lupus; however the CoQ10 analogues idebenone and MitoQ improved clinical and immunological characteristics in mouse models of lupus, suggesting a potential role for CoQ10 in the treatment of lupus patients. In the lupus-prone NZM2328 mouse model, oral supplementation with idebenone (1 g/kg for 8 weeks) reduced mortality, decreased systemic inflammation and improved renal function [67]. In lupus prone MRL-1pr mice, addition of MitoQ (200 uM) to drinking water for 11 weeks reduced oxidative stress levels and renal tissue damage [68].

## 9. Late Onset Antiphospholipid Syndrome

Antiphospholipid Syndrome (APS) is a systemic autoimmune disease which is characterized by pregnancy morbidity as well as in some cases a hyper-coagulable state of the venous or the arterial vasculature associated with the persistence of antiphospholipid antibodies [69]. The overall prevalence of APS has been estimated as 50/10,000, with age of onset typically in the range 20–50 years. However, a proportion of patients present with late onset APS, typically over the age of 65 years [70]. Mitochondrial dysfunction, oxidative stress and inflammation have been implicated in the pathogenesis of APS [71,72,73]. As an alternative to anticoagulant based therapy, research has focused on anti-inflammatory agents such as CoQ10, which in view of its antioxidant and anti-inflammatory functions, may be an appropriate candidate therapy for this disorder [1,2]. In a randomised controlled trial, patients were supplemented with the reduced form of CoQ10 (ubiquinol; 200 mg/day for 1 month); endothelial function was improved, and expression of pro-thrombotic and pro-inflammatory mediators was reduced [74]. Because of the absence of significant adverse effects, it is suggested that CoQ10 has potential as an adjunct to standard therapy in the treatment of APS.

## 10. Late Onset Lysosomal Storage Disorders

Lysosomal storage disorders (LSD) are group of more than 50 inherited metabolic disorders resulting from mutations in genes coding for individual enzymes acting within lysosomes, resulting in an accumulation of toxic metabolites [75]. This group of disorders includes mucopolysaccharidoses, mucolipidoses, Pompe’s disease, Tay-Sachs disease, Sandhoff’s disease, Niemann-Pick disease, Gaucher disease, Fabry disease, and Krabbe disease [75]. Individual disorders are rare, but the overall incidence is approximately 20/100,000 [76]. These disorders typically manifest early in life, but late onset disease variants may also occur [77]. Mitochondrial dysfunction, oxidative stress and inflammation are associated with LSDs [78,79,80]; in addition to its role in maintaining lysosomal pH, CoQ10 may also be important in lysosomal storage disorders on the basis of its role in mitochondrial function, its antioxidant role and anti-inflammatory mode of action [1,2]. A deficiency in plasma CoQ10 levels has been reported in patients with mucopolysaccharidoses [81,82] and Niemann-Pick disease [83], and supplementation with CoQ10 reportedly improves the outcomes in cell based or animal disease models of disease. In a macrophage based model of Gaucher disease, supplementation with CoQ10 partially restored mitochondrial dysfunction and decreased oxidative stress [84]. Cellular alterations present in fibroblasts from Sanfilippo disease patients were partially reversed, with a reduction in glycosaminoglycan accumulation, following administration of 50 μmol/L CoQ10 [85].

## 11. Late Onset Fibromyalgia

Fibromyalgia is a disorder characterised by both fatigue and muscle pain [86]. Fibromyalgia is associated with mitochondrial dysfunction, oxidative stress and inflammation [87,88]. Fibromyalgia is a disorder that can occur at any age in both men and women; however, fibromyalgia is thought of by medical practitioners as a condition that primarily affects middle aged women. The presence of fibromyalgia in older patients appears to have been under-investigated, and because of the likelihood of other age related problems, diagnosis of fibromyalgia in the elderly may be overlooked [89]. To date, six clinical studies assessing fibromyalgia in elderly patients have been published in the over the past 30 years. The first report of these reports was in 1988 Yunus et al. [90] who found that fibromyalgia in the elderly was often unrecognised, and treated with inappropriate medications such as steroids. The most recent study by Jacobsen et al. [91] found that more than 80% of older (55 to 95 years) patients with fibromyalgia were subject to pain, lack of mobility and sleep disruption resulting from under-treatment; in addition, many of these patients were using ineffective and potentially harmful opioid or steroid type medications. Fibromyalgia patients have been shown to have depleted tissue levels (typically 40–50% of control levels) of CoQ10 [92]. A randomised controlled clinical study by Cordero et al. [92] in 20 fibromyalgia patients found supplementation with CoQ10 (300 mg/day for 40 days) significantly reduced (by more than 50%) pain and fatigue; there was a corresponding improvement in mitochondrial energy generation, and reduced oxidative stress and inflammation. In this study, psychopathological symptoms (including depression) were also significantly improved; this was linked to the effect of supplemental CoQ10 in reducing oxidative stress and inflammation, as well as increased levels of serotonin [93,94]. In addition, Cordero et al. [95] correlated headache symptoms with reduced CoQ10 levels and increased oxidative stress in fibromyalgia patients, with headache symptoms and oxidative stress levels significantly improved following CoQ10 supplementation (300 mg/day for 3 months).

## 12. Late Onset Machado-Joseph Disease

Machado-Joseph disease is a neurodegenerative disorder, also known as spinocerebellar ataxia type 3. Clinical manifestations include ataxia, peripheral neuropathy, pyramidal signs, basal ganglia symptoms, and ophthalmoplegia, with a mean age of onset of approximately 50 years of age [96]. This disorder results from CAG (cytosine-adenine-guanine) triplet repeat expansions that encode an expanded polyglutamine tract in the disease-related protein, ataxin-3; this in turn leads to an accumulation of misfolded protein and neuronal misfunction [97]. There are currently no effective treatments for patients with this disorder. There is evidence for mitochondrial dysfunction in cellular models of Machado-Joseph disease, hence the rationale for investigating the potential benefit of CoQ10 supplementation in this disorder [98]. To date, there have been no clinical studies relating to CoQ10 supplementation in Machado-Joseph patients. However, there have been two studies supplementing CoQ10 in model systems of Machafdo-Joseph disease. In a cell model of Machado-Joseph disease, Lopes-Ramos et al. [99] described the use of CoQ10 (10–30 uM) to increase cell viability in PC-12 cells expressing expanded ataxin-3 protein. In a second study using a transgenic mouse model of Machado-Joseph diseasew, Wu et al. [100] reported supplementation with CoQ10 (1000 mg/kg/day for 7 months) reduced cerebellar degeneration and improved motor dysfunction.

## 13. Late Onset acyl-CoA Dehydrogenase Deficiency

Acyl-CoA dehydrogenase deficiency is an inherited disorder of mitochondrial fatty acid beta- oxidation, characterised by symptoms of hypoglycaemia, muscle weakness and fatigue, typically manifesting in infancy or early childhood [101]. However, it has recently become apparent that there is a late onset form of acyl-CoA dehydrogenase deficiency, in which patients first develop symptoms at a more advanced age (>60 years) [102]. Multiple acyl-CoA dehydrogenase deficiency results from a defect of electron transport from FAD-containing CoA dehydrogenases to CoQ10 in the mitochondrial electron transport chain; this in turn is due to mutations in one of the three genes, two of which encode the alpha- and beta-subunits of the electron transfer flavoprotein (*ETFA*, OMIM: 608053; *ETFB* OMIM: 130410), while the third encodes ETF ubiquinone oxidoreductase (*ETFDH* OMIM 231675). This disorder is associated with mitochondrial dysfunction, oxidative stress, inflammation as well as a deficiency in CoQ10 status in some patients [103,104,105,106]. In the study by Cornelius et al. [106], cultured fibroblasts from patients with acyl-CoA dehydrogenase deficiency had depleted levels of CoQ10 and increased levels of oxidative stress; supplementation with CoQ10 increased cellular CoQ10 levels and decreased oxidative stress. Depleted levels of CoQ10 have been reported in muscle biopsies from patients with acyl-CoA dehydrogenase deficiency, and supplementation with a combination of CoQ10 and riboflavin may result in significant symptomatic improvement [107,108]. At present the cause of the deficit in CoQ10 status associated with an acyl-CoA deficiency is uncertain, uncertain, although it may result from a transcriptional reprogramming of energy metabolism with the decrease in the level of this isoprenoid reflecting the switch from oxidative phosphorylation to aerobic glycolysis. Alternatively, it may reflect increased degradation of CoQ10 due to its aberrant binding to the mutant ETFDH protein [106].

## 14. Late Onset Leber’s Optic Neuropathy

Leber’s optic neuropathy is a hereditary disorder characterised by progressive visual loss. This disorder is caused by point mutations in the mitochondrial DNA, which results in a defect in complex I of the mitochondrial respiratory chain. This in turn results in decreased adenosine triphosphate production and increased levels of oxidative stress, causing degeneration of the retinal ganglion cells and subsequent optic atrophy [109]. Leber’s optic neuropathy typically affects young adults, but a late onset form of the disorder has been recognised in which vision loss begins in patients aged 60 or older [110]. To date there have been three randomised controlled trials supplementing the CoQ10 analogue idebenone in patients with Leber’s optic neuropathy [111,112,113]; administration of idebenone (900 mg/day) for a minimum of six months prevented further deteriation in vision. Other clinical studies have reported that administration of idebenone alone, or in combination with vitamins B2 and C, facilitated some recovery of vision in Leber’s optic neuropathy patients [114,115]. The therapeutic potential of idebenone is most likely to have the greatest impact if supplementation is initiated early in the disease, when retinal ganglion cell loss is still minimal.

## 15. Conclusions

The disorders listed above have a number of features in common, from which the following generalisations can be made: (i) These are uncommon disorders, and as such have been subject to relatively less research work and less supporting funding; (ii) For many of these disorders, either no therapy is available, or available therapies lack efficacy and/or are associated with adverse effects of concern; (iii) While the pathogenesis of many of these disorders is poorly or partially understood, mitochondrial dysfunction, oxidative stress and inflammation have been implicated in most of these disorders; (iv) on the basis of (iii) above, there is therefore a rationale for the use of supplemental CoQ10 as a novel therapeutic strategy for the management of these disorders, based on its key roles in mitochondrial function, as an antioxidant, and as an anti-inflammatory agent. The clinical trials which have assessed CoQ10 status are summarized in Table 1, together with the clinical studies which have evaluated the therapeutic efficacy of CoQ10 in the treatment of these disorders (Table 2). In summary, we have identified a number of uncommon disorders in which we consider therapeutic supplementation with CoQ10 to be warranted.

## Figures and Tables

**Table 1 antioxidants-11-02293-t001:** Clinical Trials: Coenzyme Q10 Status in Less Common Age-Related Diseases.

Author (Ref.)	Participants	CoQ10 Status	Outcome
Mitsui 2016 [15]	44 patients with MSA and 39 controls	0.51 mg/L (MSA) vs. 0.72 mg/L (controls)	Decreased levels of plasma CoQ10 in patients with MSA regardless of the COQ2 genotype
Kasai 2016 [16]	18 patients with MSA, 20 patients with Parkinson’s disease, and 18 controls	Total CoQ10 level corrected by serum cholesterol significantly lower in MSA group than in Control group	Supportive evidence for the hypothesis that decreased CoQ10 levels in brain tissue result in increased MSA risk
Du 2018 [17]	40 MSA patients 30 patients with Parkinson’s disease and 30 healthy controls	Plasma CoQ10 levels in MSA patients lower than in controls after adjusting for age, gender, total cholesterol	Correlation between decreased CoQ10 levels and the severity of motor symptoms
Compta 2018 [18]	20 patients with MSA and 15 controls	Cerebrospinal fluid levels of CoQ10 lower than in controls	Support for relevance of CoQ10 in MSA
Barca 2016 [19]	CoQ10 levels in postmortem brains of 12 MSA, 9 PD, 9 essential tremor patients, and 12 controls.	CoQ10 deficiency in MSA cerebellum associated with impaired CoQ biosynthesis and increased oxidative stress in the absence of COQ2 mutations	Evidence that CoQ10 deficiency may contribute to the pathogenesis of MSA
Schottlaender 2016 [20]	CoQ10 levels in frozen brain tissue of 20 MSA patients and 37 elderly controls	Significant decrease (by 3–5%) in the level of CoQ10 in the cerebellum of MSA cases	Suggests that a perturbation in the CoQ10 biosynthetic pathway is associated with the pathogenesis of MSA
Andrich 2004 [51]		Previously untreated HD patients (70.1 mg/L) had lower CoQ10 results than treated HD patients and controls	Evidence that that CoQ10 supplements may reduce impaired mitochondrial function in HD
Yubero 2016 [81]	Nine MPS patients	0.32 mg/L (0.20–0.53 mg/L) in eight of the nine patients	Deficiency of CoQ10 in plasma from patients with mucopolysaccha-ridoses
Montero 2019 [82]	597 individuals (average age: 11 years, range one month to 43 years)	Plasma CoQ10 significantly lower in the PKU and MPS groups than in controls and neurological patients	Plasma CoQ10 monitoring recommended to prevent chronic suboptimal blood CoQ10l status
Fu 2010 [83]	32 Niemann-Pick disease patients	Mean serum CoQ10 level of 0.48 mg/L in Niemann-Pick patients	Reduced serum CoQ10 levels associated with oxidative stress
Cordero 2012 [95]	20 fibromyalgia patients and 15 controls	Decreased CoQ10, catalase, and ATP levels in BMCs from FM patients as compared to normal controls	Association of low CoQ10 levels and increased levels lipid peroxidation in BMCs from fibromyalgia patients

**Table 2 antioxidants-11-02293-t002:** CoQ10 Supplementation Studies.

Author (Ref.)	Dosage/Duration	Participants	Outcome
Mitsui 2017 [25]	1200 mg/day for three years	Case study of individual in advanced stage of MSA	Substantial increase in total coenzyme Q10 levels CSF fluid and in plasma, clinical rating scales remained stable during the 3 years
Stamelou 2008 [32]	Liquid nano-dispersion of CoQ10 (5 mg/kg/day) or matching placebo for six weeks	21 clinically probable PSP patients (stage < or = III)	The PSP rating scale and the Frontal Assessment Battery improved slightly but significantly with the CoQ10 treatment compared to placebo, CoQ10 improved cerebral energy metabolism in PSP
Apetauerova 2016 [33]	2400 mg/day or placebo for up to 12 months	61 individuals aged 40 or older who met diagnostic criteria for PSP	No significant improvement of PSP symptoms, 41% withdrawal rate
Koroshetz 1997 [50]	360 mg/day for 2 months.	23 HD patients and 21 controls	CoQ10 treatment resulted in decreases in cortical lactate concentrations in 18 patients, which reversed following withdrawal of therapy
Huntington Study Group 2001 [54]	300 mg twice daily for 30 months	347 HD patients	No significant slowing of functional decline in early HD
McGarry 2017 [55]	2400 mg/day or matching placebo for 60 months	609 HD patients	No significant difference in patients’ Total Functional Capacity score
Perez-Sanchez 2017 [74]	200 mg/day for one month	36 patients with antiphospholipid syndrome	Improved endothelial function, reduced expression of pro-thrombotic and pro-inflammatory mediators
Fu 2010 [83]	Information on the dosage of CoQ10 or the duration of the study not provided in the manuscript	9 Niemann-Pick disease patients	Supplementation increased serum status to 1.09 mg/L but did not significantly raise the reduced CoQ10 fraction
Cordero 2013 [92]	300 mg/day or placebo for 40 days	20 fibromyalgia patients	Supplementation reduced pain and fatigue by more than 50%
Alcocer-Gomez 2014 [93]	300 mg/day or placebo for 40 days	20 fibromyalgia patients	Improvement in depression symptoms
Alcocer-Gomez 2017 [94]	300 mg/day or placebo for 40 days	20 fibromyalgia patients	Improvement in the clinical symptoms determined by FIQ questionnaire and PSQI index
Cordero 2012 [95]	300 mg/day for three months	20 fibromyalgia patients and 15 controls	Improvement in clinical symptoms and headache in fibromyalgia, increase in intracellular ATP levels in BMCs

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
