# Peer review of "Coenzyme Q10: Role in Less Common Age-Related Disorders"

_antioxidants, 2022, doi:10.3390/antiox11112293_

Round 1
Reviewer 1 Report
This review discusses the potential of using coenzyme Q10 as a therapeutic supplement for managing several less common age-related disorders.
The following suggestions are given:
1. Please confirm the title and affiliation of Iain P Hargreaves.
2. Please make a table to list all the clinical trials using coenzyme Q10 as an intervention to treat the diseases mentioned in the manuscript.
3. Except for multiple system atrophy, the mechanistic basis of using coenzyme Q10 in other diseases is less mentioned. For conditions that have not been tested with coenzyme Q10, please at least describe the result of the preclinical study that has been reported.
Author Response
- Please confirm the title and affiliation of Iain P Hargreaves.
Thank you, we have added this detail.
- Please make a table to list all the clinical trials using coenzyme Q10 as an intervention to treat the diseases mentioned in the manuscript.
Thank you, we have now added a table this all this information to the manuscript.
- Except for multiple system atrophy, the mechanistic basis of using coenzyme Q10 in other diseases is less mentioned. For conditions that have not been tested with coenzyme Q10, please at least describe the result of the preclinical study that has been reported.
Thank you, although we tried to provided this information in the previous version of the paper we have now added this detail where possible in the text, highlighted in yellow.
Reviewer 2 Report
Overall the review article is well written. It talks about the importance of CoQ10 in the less common age related disorders. They included and discussed the studies relevant to the review focus. Also, authors have mentioned the clinical trial status of CoQ10 in each disorders.
My only request is that if authors can provide the status of CoQ10 and clinical trials of each disorders in a table format also.
Author Response
Overall the review article is well written. It talks about the importance of CoQ10 in the less common age related disorders. They included and discussed the studies relevant to the review focus. Also, authors have mentioned the clinical trial status of CoQ10 in each disorders.
My only request is that if authors can provide the status of CoQ10 and clinical trials of each disorders in a table format also.
We thank the reviewer for this comment and have added 2 tables with this information.
Round 2
Reviewer 1 Report
1. The authors have made satisfactory amendments in response to our concerns.
2. The authors should also add back the missing information (words in red color) in Table 2.
Author Response
We thank the reviewer for taking the time to read our amended paper.
Regarding your comment, he authors should also add back the missing information (words in red color) in Table 2. We have now provided the missing information from the study by Koroshetz 1997, although, there is no information on the dosage of CoQ10 or the duration of the study in the manuscript by Fu, 2010 and so we have provided this information in the table, `Information on the dosage of CoQ10 or the duration of the study not provided in the manuscript.`